# The Role of Religious Coping in Caregiving Stress

Lidya Triana *[ID] and Iwan Gardono Sudjatmiko

Department of Sociology, University of Indonesia, Depok 16424, Indonesia; ig.sujatmiko09@ui.ac.id
* Correspondence: lidya.triana@ui.ac.id

**Abstract:** Studies explaining how stressors and religious coping affect caregivers' depression have been rarely conducted in the Indonesian context. Therefore, this study discusses stress process theory by examining the role of religious coping as a moderating variable between relational deprivation and loss of self on depression. In a quantitative study of 50 caregivers of persons with schizophrenia in Indonesia, this study analyzed the moderating variables using multiple regression. The results showed that higher relational deprivation will lead to increased depression, but religious coping mechanisms can reduce the effect of relational deprivation on depression (buffering effect). Religious coping can also minimize the effect of loss of self to depression. Subjective stressors and religious coping offer new theoretical insights and must be considered when studying caregiving stress. In this regard, mental health services aiming to enhance caregivers' welfare need to be provided by the state and community.

**Keywords:** stress; caregiver; religious coping; depression

## 1. Introduction

In Indonesia, the Basic Health Research in 2018 reported a prevalence of severe mental–emotional disorders (including schizophrenia) of seven individuals per thousand households (Ministry of Health of the Republic of Indonesia 2018). In Indonesia, an estimated 450,000 individuals are living with severe mental disorders. Meanwhile, the lack of accessible mental health services has led to an increase in the number of informal caregivers (family) who must care for family members suffering from schizophrenia (hereinafter, person with schizophrenia or PwS) (Keith 1995).

When accomplishing their care duties, caregivers often experience stressors and burdens (Sharif et al. 2020). A primary stressor for caregivers is PwSs' problematic behaviors as a manifestation of schizophrenia (Pearlin et al. 1990). Long-term parenting can also be a chronic stressor for caregivers (Wheaton et al. 2013). Because stress can also expand, an expansion of the primary stressor would result in role overload and role captivity (Pearlin et al. 1997). In addition, stress proliferation is more likely to occur when the primary stressor encompasses multiple social roles and relations. Caregivers' sociodemographic characteristics also introduce different stressors and seep into broader life domains such as work and family conflicts (Aneshensel 2015).

Caregivers' burdens due to care can hamper their social activities, negatively affect their family life, and induce a feeling of loss (Magliano et al. 1998). Such burdens can be generally divided into objective burden and subjective burden (Chan 2011). Objective burden is associated with disease symptoms and PwSs' behaviors including financial issues, while subjective burden includes emotional difficulties by caregivers because of their duties or PwSs' behavioral consequences (Bademli et al. 2018). A study by Tristiana et al. (2019) in Indonesia showed that, in the case of care for family members suffering from mental illness, emotional burden generated the highest score even though most caregivers also experience financial difficulties.

Depression is often linked to sociodemographic variables (gender, education level, and family dysfunction) (Cabral et al. 2014), which are also often associated with burdens

experienced by family caregivers (Chien et al. 2007; Lin et al. 2012; Bevans and Sternberg 2012). However, these difficulties can be overcome with appropriate coping mechanisms (Chang and Horrock 2006). Studies on stress processes have identified social support as more of a stress buffer resource in reducing depression (Thoits 2011; Wheaton 1985; Lin and Dean 1984) as well as a mediator of primary and secondary stressors (Streid et al. 2014). Religious coping, as a religious resource, has not been widely analyzed in studies on caregiving stress, according to Pearlin et al. (1990).

Meanwhile, the act of caregiving in Indonesia is derived from religious and spiritual values. A study by Kristanti et al. (2019) about family caregivers of people with cancer in Indonesia shows that the belief in caregiving results in care sharing among family members, sacrifices, and the use of religion and spiritualism as coping mechanisms to deal with stress due to caregiving. This is in line with Pearce (2005), who points out that caregivers have spiritual needs that are unmet in many among them, creating stress, distress, and low level of well-being. Therefore, religion becomes important and useful as an adaptive resource of treatment for caregivers.

Regarding caregiving stress, the causal relation between relational deprivation as a primary stressor, self-loss as a secondary stressor, and depression in Indonesia has not been thoroughly investigated in Pearlin et al. (1990) framework of caregiving stress. Studies about PwS caregivers' mental health in Indonesia have focused on their burdens as a result of their caregiving work, analyses of their needs, family interventions to reduce their burden, and descriptions of caregivers' and PwSs' needs in life (Tristiana et al. 2019; Jusuf 2006; Dewi 2012; Dumaria 2016). However, scholars have yet to fully examine the use of religious resources to cope with stress due to PwS caregiving in Indonesia.

Therefore, this study aims to investigate the role of religious coping as a moderating variable between stressors (relational deprivation and loss of self) and depression.

### 1.1. Caregiving and Stress Process

Caring for family members suffering from schizophrenia demonstrates a primary relationship between a caregiver and a care recipient (Pearlin et al. 1990). Working as a PwS caregiver is not easy because both PwSs and caregivers are tied to their inherent roles, be it parent–child, wife–husband, or siblings.

Schizophrenia that emerges in late adolescence/young adulthood provides a context for care (Gogtay et al. 2011), meaning that care is generally delivered by parents and can last for a long period. Long-term care of a family member with schizophrenia causes an imbalance in the relationship between the caregiver and the care recipient, as one party bears a higher burden than the other (Pearlin et al. 1990). Long-term care can also lead to the proliferation of stress (Pearlin et al. 1997), which occurs when a primary stressor in the form of an objective demand—such as PwSs' problematic behavior—results in a subjective experience associated with the tension—such as caregivers' role overload and role captivity—which ultimately results in depression. Stress escalates as demands for care increase while resources are depleted (Aneshensel et al. 1995).

#### 1.1.1. Primary Stressors

Pearlin et al. (1990) classified primary stressors into two categories: objective and subjective. The objective condition is directly linked to the worsening condition of the care recipient because of illness (Aneshensel et al. 1995). In this study, positive schizophrenic symptoms such as hallucination and delusion lead to aggressive behaviors toward oneself or others. Meanwhile, the subjective aspect is associated with the caregiver's subjective difficulties because of the care (Pearlin et al. 1990).

The primary stressor that this study analyzes is of the subjective aspect in the form of relational deprivation. Relational deprivation measures the extent to which caregivers forego experiences associated with intimacy exchange, goals, and activities (Pearlin et al. 1990). Chronic diseases such as schizophrenia have an altering effect on patients and restructure the relationship between caregivers and PwSs, dismissing the previous reciprocal rela-

tionship (Pearlin et al. 1990). As a PwS deteriorates, the caregiver experiences a sense of detachment from the part of their life that they have so far supported or shared with the PwS. Relational deprivation is often associated with caregivers caring for their spouses with dementia (Bauer et al. 2001); however, this study views relational deprivation as not only in the context of spouses, but also of parents and children, and among siblings.

### 1.1.2. Secondary Stressors

Secondary stressors also arise from care. Pearlin et al. (1990) divided secondary stressors into two parts, the first being linked to role strain and the second being associated with intrapsychic strain. This study observes intrapsychic strain, which involves dimensions of caregivers' self-concept and psychological state.

Constant difficulties by caregivers in caring for PwS can damage their self-concept and put them at risk for depression and contribute to overburden (Pearlin et al. 1981; Adams et al. 2008). One such effect on self-concept is the feeling of losing oneself. Loss of self is inextricable from the previously close relationship between caregivers and care recipients, and over time, these difficulties cause caregivers to lose themselves (Pearlin et al. 1990). Caregivers' loss of self occurs when their roles and responsibilities start to consume and exhaust their time, leaving only little room for other activities (Eifert et al. 2015). At this point, pressure builds up on a caregiver's identity as demands for care increase. In such a process, not only does an individual acquire a new role—as a caregiver—but their previous roles and identity start to disappear or become less relevant because of care responsibilities. Noonan and Tennstedt (1997) found that caregivers experience a high level of self-loss when they perform care duties at a high frequency and view caregiving as a heavy burden. Loss of self indirectly worsens depression by excessively increasing caregivers' duties, suggesting that loss of self as a result of care responsibilities can directly or indirectly affect caregivers' depressive symptoms. Meanwhile, Beeson (2003) showed that caregiver wives of persons with Alzheimer's disease experience more self-loss, loneliness, and depression than caregiver husbands. However, current studies have yet to associate loss of self with the context of PwS caregivers.

### 1.1.3. Depression

The stress process among PwS caregivers can cause mental health issues, which in this study manifest as depression. The stress process shows a causal relation, both direct and indirect (role of moderator variables), between stressors, coping mechanism, and depression (Pearlin et al. 1990).

Caregivers' excessive workloads lead to their poor health and depression (Cabral et al. 2014; Son et al. 2007). Depression and high-level stress are usually experienced by caregivers who are female, mothers, caregiving for more than five years, and have little social support (Minichil et al. 2019). This suggests that a socio-structural arrangement establishes the ground from which individual mental health problems emerge. A person's social position provides an overview of how they experience stress throughout their life.

### 1.2. Religious Coping and the Stress Process

Pearlin et al. (1990) stress process model does not view religious coping as a moderating variable that can buffer the stress effect on depression. Meanwhile, studies on stress often recognize the stress-buffering effects of one's use of personal resources such as self-esteem and mastery and social resources including social support (Pearlin et al. 1990; Aneshensel et al. 1995; Lin and Ensel 1989; Thoits 2011). On the other hand, religion can be viewed as a resource for managing mental health (Koenig 2005), helping reduce stress after adverse life events. On the one hand, positive religious coping is associated with fewer psychosomatic symptoms and an increase in one's spirituality after facing stressors (Pargament et al. 2011). It also indicates a sense of connection with transcendental powers, a secure relationship with God, and a belief that life has a greater meaning of virtue. On

the other hand, negative religious coping can moderate and hence exacerbate the stress effect on depressive symptoms (Carpenter et al. 2012).

Religious resources have advantageous effects for individuals who experience an increased level of stressful life events and conditions (Ellison and Henderson 2011). Religious resources in the forms of God Image and religious coping behavior can affect emotional and spiritual well-being for long-term cancer survivors (Gal 2000). The power of religious resources through religious coping in moderating the relationship between stress and depression varies, depending on the form of stressor (natural disaster, serious personal illness, serious illness of a loved one, violence, death of a loved one, accident, and daily hassles) and the types of subjects studied, such as undergraduate student, postgraduate student, and population in a community, not to mention the various religious backgrounds (Fabricatore et al. 2004; Khan and Watson 2006; Lee 2007; Ahles et al. 2015; Garcia et al. 2017; Gardner et al. 2014).

In addition, the forms of religious coping, such as positive religious coping and negative religious coping, have different effects as a stress buffer on depression, in which a higher level of depression happens to those who are highly exposed to stress, religious commitment, and negative religious coping (Tarakeshwar and Pargament 2001; Carpenter et al. 2012; Ahles et al. 2015). Another study that relates religious coping to caregiving also shows that negative religious coping significantly predicts depression (Herrera et al. 2009). Meanwhile, positive religious coping has been inversely related to depressive symptom in Muslim groups during the COVID-19 pandemic (Thomas and Barbato 2020).

In this study, religious resources in the form of positive religious coping (Pargament et al. 2011) are placed within the framework of stress process as a moderating variable that can reduce the impact of stress on caregiver's depression.

### 1.3. Present Research

Despite the considerable number of studies on caregiving stress, scholars have rarely examined religious coping as a moderating variable between stressors and depression, especially in the context of caregivers with family members suffering from schizophrenia.

In Indonesia, care roles are mostly performed by informal caregivers, that is, family members, amid limited mental health services. Policies remain focused on enhancing of PwSs' life quality, such as community-based rehabilitation. However, these do not specifically address caregivers' life quality (Puspitosari et al. 2019). Disease characteristics and the long duration of care contribute to caregivers' chronic stressors and depression. Caregiving demands associated with the needs of a PwS lead to subjective experiences that can undermine a caregiver's self-concept. Studies on caregiving stress have shown stress proliferation from objective to subjective (Pearlin et al. 1997). In addition, resource utilization is only associated with personal resources via self-esteem and mastery and social resources via social support (Pearlin et al. 1990; Aneshensel et al. 1995; Lin and Ensel 1989; Thoits 2011). Hence, scholars must further explore the roles of stressors, treatment resources, and depression in caregiving stress, especially for PwS caregivers. Religious resources, through religious coping, play a role in one's depressive condition, emotional well-being, and spiritual well-being to deal with harmful life events and posttraumatic symptoms (Gal 2000; Herrera et al. 2009; Fischer et al. 2010; Garcia et al. 2017).

Therefore, the following research hypotheses aim to explain the role of relational deprivation (primary stressor), loss of self (secondary stressor), and religious coping in accounting for the direct and moderating effects of stressors and depression:

**Hypothesis 1 (H1).** *Religious coping influences the relation between relational deprivation and depression.*

**Hypothesis 2 (H2).** *Religious coping influences the relation between loss of self and depression.*

Figure 1 shows the theoretical model of the study that explain the direct and moderating effect of stressors on depression.

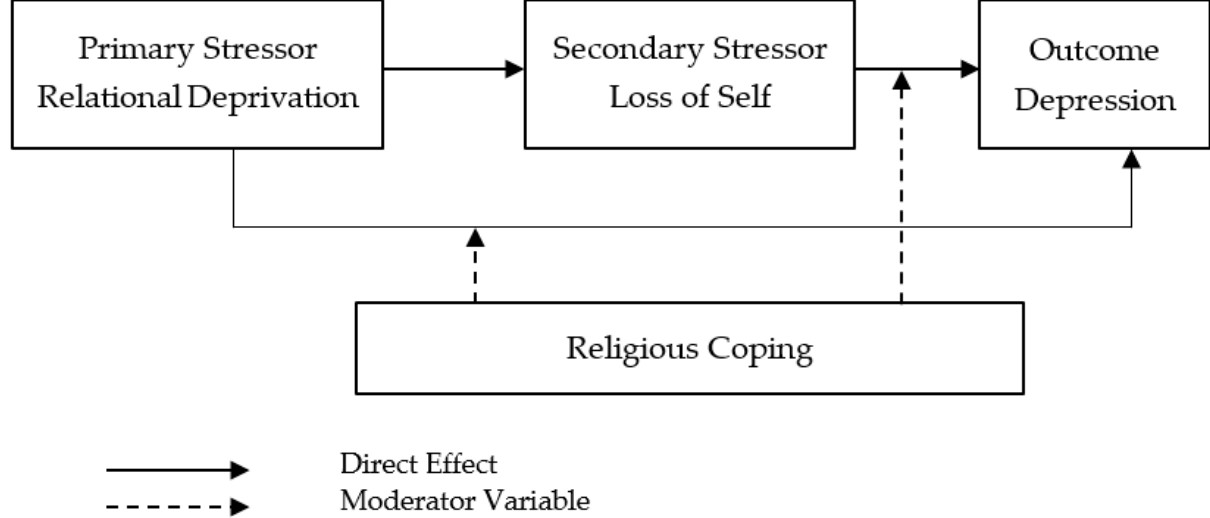

**Figure 1.** Theoretical model of study.

## 2. Method

### 2.1. Participants

A survey was administered to 50 caregivers recruited via purposive sampling, which is conducted in the absence of a sampling frame in the population. The survey population consisted of caregivers who cared for PwSs at least during the past year and were at least 18 years old. The respondents were recruited from two institutions; *Komunitas Peduli Skizofrenia Indonesia* (Indonesian Schizophrenia Care Community, 30 members) and Soeharto Heerdjan Hospital, Jakarta, Indonesia (20 daycare participants).

Among the respondents, 60% were parents, 26% siblings, 8% spouses, and 6% children. Table 1 shows that female caregivers outnumbered male caregivers. Meanwhile, in terms of age, caregivers aged 60–69 constituted the largest group (58%).

**Table 1.** Caregiver characteristics (*n* = 50). PwS, person with schizophrenia.

| Characteristics | *n* | % | Mean of Depression Score | Mean of Relational Deprivation Score | Mean of Loss of Self Score | Mean of Religious Coping Score |
|---|---|---|---|---|---|---|
| **Relationship with PwS** | | | | | | |
| Parent | 30 | 60 | 13.2 | 11.9 | 4.4 | 24.5 |
| Wife/husband | 4 | 8 | 6.5 | 14.2 | 3.0 | 28.0 |
| Child | 3 | 6 | 13 | 17.3 | 4.7 | 25.0 |
| Sibling | 13 | 26 | 12.7 | 12.8 | 3.3 | 23.9 |
| **Sex** | | | | | | |
| Male | 16 | 32 | 11.2 | 11.8 | 4.4 | 25.4 |
| Female | 34 | 66 | 13.1 | 13.0 | 3.8 | 24.3 |
| **Religion** | | | | | | |
| Islam | 39 | 78 | 12.4 | 12.5 | 4.3 | 25.1 |
| Christian | 6 | 12 | 16.0 | 15.7 | 3.5 | 24.7 |
| Catholic | 5 | 10 | 9.4 | 9.8 | 2.8 | 21.4 |
| **Age** | | | | | | |
| 20–29 | 4 | 8 | 13.5 | 13.5 | 2.5 | 20.5 |
| 30–39 | 5 | 10 | 15.6 | 12.2 | 4.0 | 26.2 |
| 40–49 | 9 | 18 | 11.9 | 14.8 | 4.1 | 25.1 |
| 50–59 | 11 | 22 | 11.0 | 11.6 | 3.6 | 24.7 |
| 60–69 | 19 | 58 | 13.5 | 12.6 | 4.8 | 24.6 |
| 70–79 | 2 | 4 | 5.5 | 8.0 | 2.0 | 28.0 |

Table 1 also shows the mean scores of this study's variables: depression, relational deprivation, loss of self, and religious coping. For depression, the highest mean scores can be seen in parent caregivers (13.2), female (13.1), Christian (16.0), and within the 30–39 age group (15.6). Meanwhile, for relational deprivation, the highest mean scores were among child caregivers (17.3), female (13.0), Christian (15.7), and within the 40–49 age group (14.8). For loss of self, the highest mean scores can be seen in child caregivers (4.7), male (4.4), Islam (4.3), and within the 60–69 age group (4.8). Lastly, for religious coping, the highest mean scores were found in spouse caregivers (husband/wife) (28.0), male (25.4), Islam (25.1), and within the 70–79 age group (28).

### 2.2. Procedure

To begin data collection, formal and informal contacts were made with administrators of *Komunitas Peduli Skizofrenia Indonesia* and the daycare unit of Soeharto Heerdjan Hospital. After obtaining approval, the researchers conducted a questionnaire survey to caregivers who agreed to be research subjects, who were then given informed consent forms to read and understand. The caregivers themselves filled out the questionnaires (self-administered). Data collection took place from August to October 2018.

### 2.3. Instruments

Relational deprivation variable is measured through two dimensions, i.e., deprivation of intimacy exchange and deprivation of goal and activity (Pearlin et al. 1990). Deprivation of intimacy exchange is measured from feelings of missing the formerly known person (PwS) and being able to confide in the PwS as well as losing someone who is close to the caregiver. Meanwhile, goal and activity deprivation is linked to caregivers' feelings of loss regarding practical activities usually done with the PwS and their loss of contact with other people. Each dimension consists of three indicators measured in the Likert scale of 1–4, where 1 = "not at all" and 4 = "completely". Reliability test results for this variable showed a Cronbach's $\alpha$ of 0.903.

Based on Pearlin et al.'s (1990) proposed loss-of-self indicator, the caregivers reported the extent to which they feel they have missed out on important things in life because of schizophrenia. This indicator includes the caregiver's loss of identity and an important part of themselves. This loss of self can happen completely, substantially, somewhat, or not at all (1–4 point of Likert scale). Reliability analysis showed a Cronbach's $\alpha$ of 0.941.

Meanwhile, this study measured religious coping by adopting Pargament et al. (2011) Brief RCOPE for positive religious coping (PRC). The study of Pargament et al. (2011). demonstrates that there are efforts to use the Brief RCOPE in some religious and cultural settings, and this constitutes an early effort to validate the Brief RCOPE among varied religious and cultural groups, including the context of Islam (Khan and Watson 2006). An individual performs positive religious coping by seeking a stronger relationship with God, his love and attention, and his help to release anger; carrying out plans with him; trying to see how he strengthens caregivers in difficult situations; asking for his forgiveness for their sins; and focusing on religion to stop worrying about problems (Pargament et al. 2011). PRC items are answered using 1–4 points of a Likert scale, where 1 = "not at all" and 4 = "great deal". Reliability analysis for this variable showed a Cronbach's $\alpha$ of 0.923.

To measure depression, this study used CES-D (The Center for Epidemiologic Studies Depression) Scale (Radloff 1977). The tool consists of 20 questions measuring a person's depression, scored between 0 and 3 for each item. The total score ranges from 0 to 60; the higher the score, the more depression symptoms there exist. Reliability analysis for this variable showed a Cronbach's $\alpha$ of 0.834.

### 2.4. Data Analysis

Descriptive statistics is used to demonstrate the mean score of positive religious coping subscale items. Meanwhile, Pearson correlation was conducted on all study variables, while multiple regression analyses were carried out to test the moderating variables.

The moderating variable was analyzed via hierarchical multiple regression to examine the interaction effects between relational deprivation and positive religious coping in predicting caregivers' depression (Hayes 2013). The same analysis was used to determine the interaction effects between loss of self and positive religious coping in predicting caregivers' depression. A scatterplot was generated to show how positive religious coping affects the relation between relational deprivation and depression and the relation between loss of self and depression. The overall analysis of the moderator variables used IBM SPSS Statistics 22.

## 3. Results

The mean score of positive religious coping can be seen from the seven question items based on the Brief RCOPE developed by Pargament et al. (2011). Table 2 shows the mean score of positive religious coping subscale items among three religions.

**Table 2.** Mean score of positive religious coping subscale items among three religions (*n* = 50).

| No | Items | Islamic M (SD) | Christian M (SD) | Catholic M (SD) |
|----|-------|------------------|--------------------|-------------------|
| 1 | Looked for a stronger connection with God. | 3.61 (0.59) | 3.50 (0.55) | 3.20 (0.84) |
| 2 | Sought God's love and care. | 3.59 (0.64) | 3.50 (0.55) | 2.60 (1.14) |
| 3 | Sought help from God in letting go my anger. | 3.59 (0.64) | 3.67 (0.52) | 3.00 (0.71) |
| 4 | Try to put my plans into action together with God. | 3.54 (0.60) | 3.50 (0.55) | 3.20 (0.84) |
| 5 | Tried to see how God might be trying to strengthen me in this situation | 3.59 (0.68) | 3.33 (0.52) | 3.20 (0.84) |
| 6 | Asked forgiveness for my sins | 3.72 (0.51) | 3.67 (0.52) | 3.20 (0.84) |
| 7 | Focused on religion to stop worrying about my problems | 3.46 (0.82) | 3.50 (0.55) | 3.00 (0.71) |

The mean score of PRC indicates that Muslim caregivers have a higher score of PRC compared with the Christian and Catholic caregivers in 5 of 7 question items. Meanwhile, Catholic caregivers have the lowest mean score compared with the others for all PRC items. The item that has the highest score in all caregivers concerns asking for God's mercy and forgiveness for all of one's sins (Muslim = 3.72, Christian = 3.67, Catholic = 3.20). In Pargament et al. (2011), this is categorized as religious purification in the dimension of religious methods of coping to gain control and closeness to God. To Muslim caregivers, the second highest score is looking for a stronger connection with God (3.61), which shows the aspect of spiritual connection. Meanwhile, in Christian caregivers, the other high score is seeking help from God in letting go of my anger (3.67), which illustrates the aspect of religious forgiveness.

Table 3 shows the result of the Pearson correlation test on all of the analyzed variables in which relational deprivation, loss of self, and positive religious coping have a significant correlation with depression, while relational deprivation has a significant correlation with loss of self.

**Table 3.** Descriptive statistics and Pearson's correlation analysis (*n* = 50).

| Variables | *M* | SD | 1 | 2 | 3 | 4 |
|-----------|-----|-----|---|---|---|---|
| Relational Deprivation | 12.6 | 4.67 | 1 | 0.351 * | 0.016 | 0.303 * |
| Loss of Self | 4.04 | 2.04 | | 1 | 0.129 | 0.365 ** |
| Religious Coping | 24.7 | 3.85 | | | 1 | −0.249 *** |
| Depression | 12.5 | 9.26 | | | | 1 |

Note: * $p < 0.05$, ** $p < 0.01$, *** $p < 0.1$.

Further analysis was performed on positive religious coping, which plays a moderating role not only between relational deprivation and depression, but also between self-loss and depression. A separate analysis was conducted on positive religious coping as a moderating variable between relational deprivation and depression, which continued

with an analysis of self-loss and depression. Both were performed via hierarchical multiple regression consisting of two equations.

$$Y = \text{intercept} + b_1 X + b_2 M \tag{1}$$

$$Y = \text{intercept} + b_1 X + b_2 M + b_3 X \times M \tag{2}$$

Table 4 presents the hierarchical multiple regression analysis of the interaction effect between relational deprivation and positive religious coping in predicting depression.

**Table 4.** Analysis of the moderating role of religious coping between relational deprivation and depression.

| Predictor | | Coeff B [a] | SE B | B [b] | p | R2 |
|---|---|---|---|---|---|---|
| Intercept | $i_1$ | 12.540 | 1.228 | | 0.000 | 0.157 |
| Relational Deprivation | $b_1$ | 0.610 | 0.266 | 0.307 * | 0.026 | |
| Positive Religious Coping | $b_2$ | −0.611 | 0.322 | −0.254 *** | 0.064 | |
| | | | | | | |
| Intercept | $i_2$ | 12.581 | 1.197 | | 0.000 | 0.217 |
| Relational Deprivation | $b_1$ | 0.715 | 0.265 | 0.360 * | 0.010 | |
| Positive Religious Coping | $b_2$ | −0.762 | 0.324 | −0.317 * | 0.023 | |
| Relational Deprivation * Positive Religious Coping | $b_3$ | −0.144 | 0.077 | −0.258 *** | 0.067 | |

Note: * $p < 0.05$, *** $p < 0.1$, a = unstandardized coefficient, b = standardized coefficient.

Table 4 shows a statistically significant interaction between relational deprivation and positive religious coping, indicating that the positive religious coping mechanism moderates the relation between relational deprivation and depression. However, while increased relational deprivation will lead to increased depression, positive religious coping can reduce the effect of relational deprivation on depression (buffering effect).

Hierarchical multiple regression analysis was also performed to examine the interaction effect between loss of self and positive religious coping in predicting depression (Table 5).

**Table 5.** Analysis of the moderating role of positive religious coping between loss of self and depression.

| Predictor | | Coeff B [a] | SE B | B [b] | *p* | $R^2$ |
|---|---|---|---|---|---|---|
| Intercept | $i_1$ | 12.540 | 1.179 | | 0.000 | 0.222 |
| Loss of Self | $b_1$ | 1.833 | 0.589 | 0.404 ** | 0.003 | |
| Positive Religious Coping | $b_2$ | −0.724 | 0.312 | −0.301 * | 0.025 | |
| | | | | | | |
| Intercept | $i_2$ | 12.843 | 1.168 | | 0.000 | 0.270 |
| Loss of Self | $b_1$ | 2.197 | 0.614 | 0.484 ** | 0.001 | |
| Positive Religious Coping | $b_2$ | −0.959 | 0.334 | −0.399 ** | 0.006 | |
| Loss of Self * Positive Religious Coping | $b_3$ | −0.306 | 0.176 | −0.248 *** | 0.090 | |

Note: * $p < 0.05$, ** $p < 0.01$, *** $p < 0.1$, a = unstandardized coefficient, b = standardized coefficient.

Multiple analyses of the interaction effects of loss of self and positive religious coping on depression suggest statistically significant results ($\beta = -0.248$), indicating that depression associated with loss of self depends on the extent of positive religious coping. Caregivers with a high sense of loss experienced increased depression, but positive religious coping was found to reduce the impact of such loss on depression.

Figure 2 shows the scatterplot of the effect of religious coping on the relation between relational deprivation and depression and between loss of self and depression.

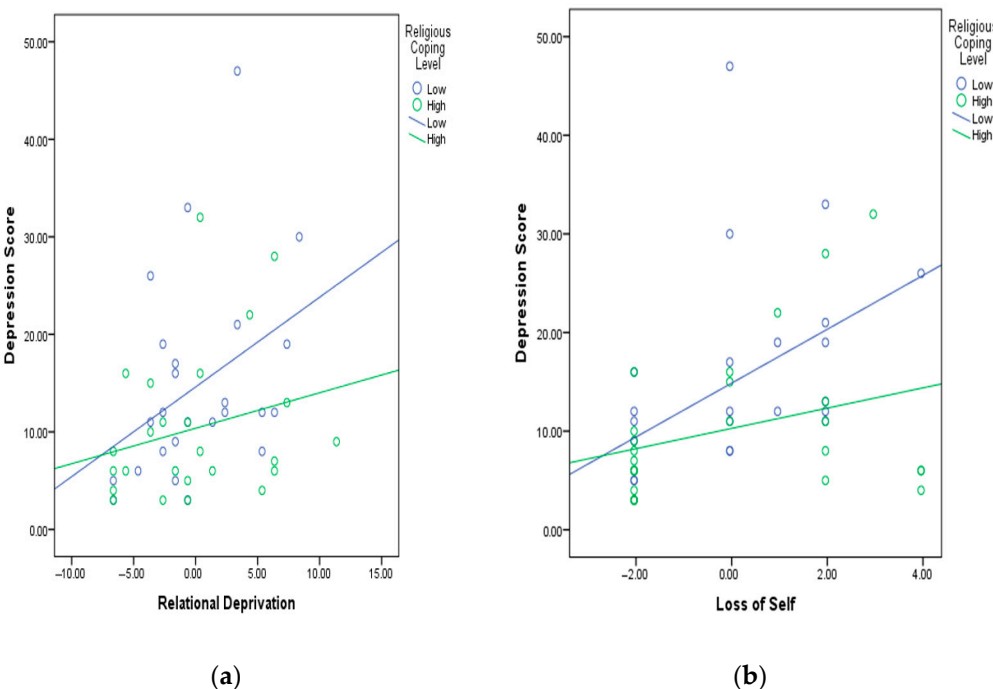

(**a**)                                           (**b**)

**Figure 2.** (**a**) Effect of religious coping on the relation between relational deprivation and depression; (**b**) effect of religious coping on the relation between loss of self and depression.

## 4. Discussion

The analysis of the correlation between relational deprivation and loss of self suggests that a restructuring of the relationship between caregivers and PwSs, compared with before the PwSs developed the illness, produced a more unidirectional relationship with no reciprocity, which then leads to the caregivers' loss of self. Caregivers' feeling of losing their self-identity, as their relationship with PwSs is no longer the same, can increase their depression; their burdens increase with care duties taking up most of their activities, leaving little room for personal pursuits, finally leading to their loss of self-identity (Adams et al. 2008; Eifert et al. 2015; Noonan and Tennstedt 1997). This study demonstrates that relational deprivation and loss of self as stressors have a high correlation, i.e., the higher the relational deprivation experienced by a caregiver, the higher the caregiver's loss of self as well. Meanwhile, both relational deprivation and loss of self as stressors have a positive correlation with depression, in which a high level of stressor results in a high level of depression as well. Subjective stressor influences the output of a caregiver's mental health more strongly.

Meanwhile, the analysis of the moderating effect of religious coping suggests that increased relational deprivation will lead to increased depression, but that positive religious coping can buffer the effect of such deprivation on depression. The reciprocal relationship of a caregiver and PwS before the illness of the latter fades away as the PwS's symptoms escalate. In addition, the caregiver also simultaneously grows apart from activities beyond the role of caregiving, all resulting in relational deprivation (Pearlin et al. 1990), although positive religious coping can reduce its impact on depression. The image of positive religious coping that is most often used by all respondents, whether Muslim, Christian, or Catholic in this study, is religious purification. Another aspect often carried out in coping is spiritual connection by Muslim respondents and religious forgiveness by Christian respondents. Religious purification and spiritual connection are carried out to attain comfort and closeness to God, while religious forgiveness is done to attain life transformation (Pargament et al. 2011).

Further analysis of how religious coping affects the relation between loss of self and depression suggests similar results. Caregivers with a high sense of loss will get

more depressed, but positive religious coping can reduce the impact of such loss on depression. Loss of self, as a result of care responsibilities, occurs because caregivers lose their identity and tend to negate their previous activities and roles even though they confirm their identity through such activities (Pearlin et al. 1990), which can directly affect their depression (Adams et al. 2008). Studies have mostly found a causal relation between the direct effect of loss of self and depression. This study strengthens the literature on the importance of positive religious coping in reducing the impact of loss of self on depression, also known as stress buffering model (Ellison and Henderson 2011). This model in this study can also be seen from the role of positive religious coping in reducing the impact of relational deprivation on a PwS caregiver's depression.

Studies explaining the role of religious coping have shown its direct effect on depression, emotional and spiritual well-being, and the ability to overcome adverse life events (Gal 2000; Herrera et al. 2009; Gardner et al. 2014; Thomas and Barbato 2020). Meanwhile, studies examining religious coping as a moderating variable have suggested varying results, such as the roles of positive and negative religious coping on posttraumatic symptoms and depression (Garcia et al. 2017; Carpenter et al. 2012), the mitigating effect of religious/spiritual coping on the impact of stress on students' depression (Lee 2007), the moderating role of negative religious coping between stress and depression among students in Christian universities (Ahles et al. 2015), and the moderating role of deferring religious coping between stress and well-being among undergraduate students in religiously affiliated university (Fabricatore et al. 2004).

Indonesia, home to the largest number of Muslims in the world (Pew Research Center 2015) has a strong force and broad scope of Islamic values, which include daily social interactions, relationships, and religious practices (Sudjatmiko et al. 2018a). Islamic religious practices and values can also increase happiness and reduce suicide rates (Sudjatmiko et al. 2018b). Thomas and Barbato (2020), who studied how individuals face adverse life experiences, showed that Muslims show higher religious coping than Christians, which is associated with a higher level of religiosity expressed through religious practices such as daily prayer and visiting places of worship. In another study on Muslim students in Pakistan, it is demonstrated that positive religious coping correlates negatively with depression albeit after the negative religious coping in the analysis is taken out (Khan and Watson 2006). Meanwhile, in the context of caregiving, the experiences of family caregivers of people with cancer in Indonesia show that the belief in caregiving is the center of the phenomenon that leads to acts of care-sharing, mutual sacrifice, and coping mechanism based on religious and spiritual values, in which the use of positive religious coping delivers high satisfaction to caregivers compared with those who do not use it (Kristanti et al. 2019). Another study by Tarakeshwar and Pargament (2001) confirms that positive religious coping is related to better religious outcome and greater stress-related growth, but unrelated to anxiety and depression.

The result of the study presents a significant role of positive religious coping as a moderating variable in reducing a stressor's impact on caregivers' depression. However, this study has its limitation, which is the use of positive religious coping that is only limited to caregivers of PwSs. The clinical implication is the proposition to consider an expansion of coping resources, not only covering personal resources (improving self-esteem and mastery) and social support through a network of support system, but also religious coping that can reduce the stress impact as a result of caregiving, especially in informal caregivers. It is necessary for professional helpers to consider aspects of religious and spiritual coping as essential resources needed by caregivers in terms of intervention (Lee 2007). The use of positive religious coping can be done by facilitating caregivers within the setting of health services by primary service provider (Herrera et al. 2009) or the state's and community's support through various kinds of platforms, both online (WhatsApp group, Facebook, YouTube, apps, and so on) and offline by implementing the Covid-19 health protocol.

## 5. Theoretical Conclusions

These findings show three important phenomena. First, caregiving duties can be detrimental to a caregiver's health, which corresponds to depression in this study. Caregivers' stress process suggests that primary stressors can affect secondary stressors and affect their depression. Relational deprivation, as a primary stressor, can affect a caregiver's loss of self, increasing their depressive symptoms. The subjective aspects of primary and secondary stressors are more dominant than those of other stressors.

Second, no other studies have explained the role of religious coping in the context of PwS care. Therefore, this study extends Pearlin's model by integrating religion and strengthens the assertion that religious coping reduces the impact of relational deprivation and loss of self on depression among caregivers of PwSs. Third, the study extends and reconfirms Pargament's construct of positive religious coping to caregivers PwS with different religions in Indonesia. Nevertheless, future studies on caregiving stress would benefit from investigating the role of different religions to provide a more comprehensive picture of the impact of religion.

**Author Contributions:** Conceptualization, methodology, formal analysis, L.T.; writing—original draft preparation, L.T.; writing—review and editing, L.T. and I.G.S.; theoretization, L.T. and I.G.S.; supervision, I.G.S. All authors have read and agreed to the published version of the manuscript.

**Funding:** This research was funded by *Lembaga Pengelola Dana Pendidikan* (LPDP) Republik Indonesia grant number: FR3072018148443, year 2018.

**Institutional Review Board Statement:** Ethical review and approval were waived for this study, due to this research poses minimal risk for participants because they only respond to questions through a questionnaire without further investigation.

**Informed Consent Statement:** Informed consent was obtained from all subjects involved in the study.

**Data Availability Statement:** The data presented in this study are available on request from the corresponding author. The data are not publicly available due to privacy of the participants.

**Conflicts of Interest:** The authors declare no conflict of interest.

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
