# Peer review of "The Role of Religious Coping in Caregiving Stress"

_religions, doi:10.3390/rel12060440_

Round 1

Reviewer 1 Report

Generally a well written paper 

The introductory literature review is fairly comprehensive but there is a need for more general discussion of the literature on religious coping and caregiving . The methodology and analysis are sound. 

  1. The authors should discuss how cultural factors in Indonesia impact caregiving and also religious coping ie how religious backgrounds impact such coping
  2. Is the R COPE culturally validated
  3. What are the clinical implications of the findings?
  4. Could the authors define relational deprivation? 

Reviewer 2 Report

The study of 50 caregivers of patients with schizophrenia in Indonesia, which analyzed these mediating and moderating variables using multiple regressions. The results showed that loss of self mediated the relation between relational deprivation and caregiver depression.

  1. The sample size=50 may not be enough for mediation analysis. I suggest to omit this part or increasing sample size (please refer the medication analysis sample size package, such as R-package--> powerMediation)
  2. Regressions with 2 independent variables is just fit for power analysis, thus tables 3 and 4 could be kept in the article. However, what difference between the Coeff B and β? Is the β standardized regression coefficient? if so, please note in the table footnote and text.
  3. It is recommended to describe the religious coping mechanism and the differences between different religions in detail.
  4. The research object is schizophrenia, which is not representative to all cases. After all, the coping style of schizophrenia is different from ordinary people. Please discuss this point as the limitation of the article.
  5. The authors suggest that mental health services need to be provided at the community level. However, now that COVID-19 is severe, it is recommended to develop suitable support systems and models for the needs of the society at this stage.
  6. The reasons for the loss of each case are different, mainly affecting the caregiver stress. For example, cancer, suicide, COVID-19... Is there correction?

Round 2

Reviewer 2 Report

I have no more comments.

This manuscript is a resubmission of an earlier submission. The following is a list of the peer review reports and author responses from that submission.